# Identifying QTLs Related to Grain Filling Using a Doubled Haploid Rice (*Oryza sativa* L.) Population

So-Myeong Lee [1,†], Nkulu Rolly Kabange [1,†], Ju-Won Kang [1], Youngho Kwon [1], Jin-Kyung Cha [1], Hyeonjin Park [1], Ki-Won Oh [1], Jeonghwan Seo [2], Hee-Jong Koh [3] and Jong-Hee Lee [1,*]

1 Department of Southern Area Crop Science, National Institute of Crop Science, RDA, Miryang 50424, Republic of Korea
2 Crop Breeding Division, National Institute of Crop Science, RDA, Jeonju 55365, Republic of Korea
3 Department of Plant Science, Seoul National University, Seoul 08826, Republic of Korea
* Correspondence: ccriljh@korea.kr; Tel.: +82-53-350-1168; Fax: +82-55-352-3059
† These authors contributed equally to this work.

**Abstract:** Grain filling is an important trait of rice that affects the yield of grain-oriented crop species with sink capacity-related traits. Here, we used a doubled haploid (DH) population derived from a cross between 93-11 (P1, *indica*) and Milyang352 (P2, *japonica*) to investigate quantitative traits loci (QTLs) controlling grain filling in rice employing the Kompetitive allele-specific PCR (KASP) markers. The mapping population was grown under early-, normal-, and late-cultivation periods. The phenotypic evaluation revealed that spikelet number per panicle positively correlated with the grain-filling ratio under early cultivation conditions. Notably, three significant QTLs associated with the control of grain filling, *qFG3*, *qFG5-1*, and *qFG5-2*, were identified. Genes harbored by these QTLs are linked with diverse biological processes and molecular functions. Likewise, genes associated with abiotic stress response and transcription factor activity and redox homeostasis were detected. Genes such as *MYB*, *WRKY60*, and *OsSh1* encoding transcription factor, β-catenin, and the tubulin FtsZ, as well as those encoding cytochrome P450, would play a forefront role in controlling grain filling under early cultivation conditions. Our results suggest that *qFG3*-related genes could mediate the transition between grain filling and abiotic stress response mechanisms. Fine-mapping these QTLs would help identify putative candidate genes for downstream functional characterization.

**Keywords:** grain filling; yield; quantitative trait locus; rice





## 1. Introduction

Rice is a staple food crop feeding nearly 50% of the global population. The improved field management and genetics of rice have brought breakthroughs in rice yield over the past decades [1–3]. Nevertheless, developing rice varieties with a high-yielding potential is still at the center of interest in many breeding programs [4], which could be partly explained by the increasing global population and emerging environmental challenges, employing various breeding methods [3,5–7]. Of the various traits associated with rice yield, grain filling is regarded as a limiting factor for rice productivity and quality. Currently, many rice breeders are trying to improve rice yield by developing 'super rice' cultivars that have high-yield potential due to the large spikelet number per panicle; however, their poor grain filling of inferior spikelets renders the goal unattainable. The filling ratio of the rice grain is the final factor determining rice yield, which is closely associated with the sink and source ability. Moreover, further studies are still needed to address the problem of breeding rice cultivars with high-yield rice cultivars and high grain-filling ratios in molecular approaches [8,9].

Several reports support that grain filling and starch metabolism in plants are tightly linked [10–12]. Similarly, an efficient grain-filling rate favors a high grain weight [8,13]. The starch metabolism in rice has been comprehensively investigated [11,14–20]. As per some

evidence, rice sink strength and grain filling have an antagonistic relationship. According to Chen et al. [21], rice plants with large panicles have a large sink capacity, while exhibiting a slow and low spikelet filling, especially inferior spikelets in the panicle. This is ultimately due to the asynchronous grain filling, in which sucrose synthase (SUS), among other enzymes, and abscisic acid (ABA) are said to play an important role [10]. Efforts toward increasing rice yield through the enlargement of sink size or strength often result in sink-source imbalance, leading to a decrease in the source productivity or grain filling of the rice grain [22].

Nagata et al. [23] reported that the *indica* allele at a QTL associated with sink size and ripening in rice, mapped on chromosomes 1 and 6, improved rice yield by increasing the number of primary and secondary rachis without causing any apparent decrease in grain filling. Under these conditions, maintaining sink productivity even when the number of spikelets per panicle increased would be more favorable. According to San et al. [24], a near-isogenic line (NIL derived from a cross between the *indica* cultivar Takanari and *japonica* cultivar Koshihikari) carrying the *indica* allele at *qLIA3* showed more tilted leaves than Koshihikari, with the latter showing a greater photosynthesis efficiency at the ripening stage. In addition to the factors mentioned above, nitrogen use efficiency (NUE) has proven to be a crucial factor in sink productivity. In this regard, Yamamoto et al. [25] reported that *qCHR1* associated with nitrogen transport mapped on Chr 1 improved nitrogen accumulation and distribution to leaves during ripening, but no change in the yield was observed. Meanwhile, an RIL carrying the *japonica* allele at *LSCHL4*, a gene reported in the narrowed-leaf mutant of rice, recorded a nearly 18.7% increase in yield higher than the *indica* counterpart. The *japonica* allele at *LSCHL4* conferred a high leaf chlorophyll content, improved panicle type, and enlarged flag leaf size [26].

This study aimed to investigate genetic regions controlling high-grain filling rice under early cultivation conditions. To achieve that, a doubled haploid (DH) population was used along with Kompetitive allele-specific PCR (KASP) markers and Fluidigm markers. Several QTLs associated with grain filling were detected, of which three QTLs (*qFG3*, *qFG5-1*, and *qFG5-2*) with high LOD scores were identified as significant in the early cultivation environment.

## 2. Materials and Methods

### 2.1. Plant Materials and Growth Conditions

A total of 117 DH lines developed through anther culture, derived from an F1 plant crossed between the typical *indica* rice cultivar 93-11 (a high-yielding cultivar originated from China) and the *japonica* cultivar Milyang352 (a Korean cultivar with moderate yield and early maturity), was used to conduct the experiments. The population was developed in summer 2017 in Miryang, Korea. Prior to transplanting, seeds were surface sterilized with a diluted chemical solution (copper hydroxide 0.012%, Ipconazole 0.016%, 32 °C) and soaked for 48 h to induce germination. Germinated seeds were sown and grown for about 3 weeks in 50-well trays, followed by transplanting in an open field in Miryang, Korea. Each DH line was transplanted in order of entry no. of each line, once for each cultivation season in 4 rows. The row length was 4 m, the hills within the row were spaced by 15 cm (26 plants per each line), and the space between rows was 30 cm.

The experiments were conducted during three consecutive years, with two cultivation seasons in 2018 and one cultivation season in 2019 and 2021. During the first year (2018), the mapping population was cultivated in early transplanting (first half of May) and late transplanting (first half of July) under a normal nitrogen cultivation regime. For the last two years, 2019 and 2021, plants were grown during the regular rice cultivation season in Korea (first half of June).

### 2.2. Phenotypic Measurements

To assess the grain-filling ratio, rice panicles from each plant were harvested 45 days after heading in triplicate, and threshed manually in a plastic box to prevent loss of spikelets.

Then, the filled grains in this study were defined as grains sinking into the NaCl solution (2.7 *M*) [27]. The filled grain ratio (FG) was calculated using the following formula, FG = [(number of filled grains/total number of grains per panicle) × 100], and was expressed in percentage. Other agronomic traits were also investigated: thousands grain weight (TGW = [(average filled grain weight/number of samples) × 100]), spikelet number per panicle (SN), and panicle number (PN) per plant. Flowering time was determined by the date when 40–50% of panicles emerged in a line. All samples were collected from the inside rows, excluding the border rows, to avoid the border effects on the traits studied or competition between lines. A correlation analysis between the grain-filling ratio and other traits was performed by analysis protocol in GraphPad Prism 7.00. A GxE analysis of FG was conducted by R (version 4.2.2) and "which-won-where" view of the GGE biplot were made by R Studio package "ggplot2".

### 2.3. DNA Extraction and Molecular Markers Analysis

The genomic DNA was extracted from leaf samples using the CTAB method [28], with slight modifications. In essence, snap-frozen samples (with liquid nitrogen for a few seconds) were ground to a fine powder in 2 mL Eppendorf (Eppendorf, T2795) tubes with 3 mm stainless steel beads (Masuda, Cat. 12-410-032, Tokyo, Japan). Next, 700 μL of DNA Extraction buffer (D2026, Lot D2622Z12H, Biosesang, Seongnam-si, Korea) was added to the ground leaf sample in 2 mL tube, and the mixture was vortexed, followed by incubation at 65 °C for 30 min in a dry oven. Then, 500 μL PCI solution (phenol:chloroform:isoamylalcohol = 25:24:1, Sigma-Aldrich, St. Louis, MO, USA) was added, followed by gentle mixing by inversion and centrifugation for 15 min at 13,000 rpm. The supernatant (500 μL) was transferred to fresh 1.5 mL tubes, isopropanol (500 μL) was added, and the mixture was incubated for 1 h at −20 °C. Soon after, tubes were centrifuged at 13,000 rpm to pellet down the DNA. The pellets were washed by adding 70% ethanol (EtOH, 1 mL), the tubes were centrifuged at 13,000 rpm for 10 min, and EtOH was discarded. DNA pellets were dried at room temperature and later resuspended in 100 μL 1× TE buffer (10 mM Tris-HCL, pH 8.0; 2.5 mM EDTA). DNA samples were quantified using nano-drop (ND1000 spectrophotometer, Mettler Toledo, Greifensee, Switzerland).

For the construction of the molecular map, KASP marker amplification and allelic discrimination were performed using the Nexar system (LGC Douglas Scientific, Alexandria, VA, USA) at the Seed Industry Promotion Center (Gimje, Republic of Korea) of the Foundation of Agri. Tech. Commercialization & Transfer in Korea. An aliquot (0.8 μL) of 2X master mix (LGC Genomics, London, UK), 0.02 μL of 106 KASP assay mix (LGC Genomics), containing one KASP SNP marker in each mix, and 5 ng genomic DNA template were mixed in a 1.6 μL KASP reaction mixture in a 384-well array tape. KASP amplification was performed as described by Cheon et al. [29].

A total of 240 SNP markers, including 106 KASP SNP markers and 134 Fluidigm SNP markers, were used for genotyping the DH population, and genotype data of 230 polymorphic SNP markers were used for final mapping (Figure S1, Table S1–S3) [29,30]. Fluidigm markers for SNP genotypes were determined using the BioMark™ HD system (Fluidigm, San Francisco, CA, USA) and 96.96 Dynamic Array IFC (96.96 IFC) chip according to the manufacturer's instructions at the National Instrumentation Center for Environmental Management (NICEM), Seoul National University (Pyeongchang, Republic of Korea). The genotyping results were obtained using the Fluidigm SNP Genotyping Analysis software. All genotype calls were manually confirmed, and any errors in homozygous or heterozygous clusters were curated.

### 2.4. Linkage Mapping and QTL Analysis

The genotype and phenotype data, comprising 230 polymorphic SNP markers (96 KASP markers and 134 Fluidigm markers) and 117 DH lines, were used to perform linkage mapping and QTL analysis with IciMapping software v.4.1 for a bi-parental population (position mapping and Kosambi mapping functions were selected) in order to detect genetic

loci associated with grain-filling traits in rice. The logarithm of the odds (LOD) threshold 3.0 was selected for detecting significant QTLs based on Akond et al. [31]. The proportion of observed phenotypic variance explained by each QTL and the corresponding additive effects were also estimated.

*2.5. Gene Ontology Search of qFG3, qFG5-1, and qFG5-2-Related Genes*

The genome browser in the rice genome annotation project database (http://rice.uga.edu/cgi-bin/gbrowse/rice/, accessed on 29 August 2022) and the PlantPAN 3.0 database (http://plantpan.itps.ncku.edu.tw/search.php#species, accessed on 29 August 2022) were used to identify the locus ID and the basic annotation of the genes within the QTL region. The Plant Transcription Factor Database (http://planttfdb.gao-lab.org/, accessed on 29 August 2022) was used to verify whether a particular gene encodes a transcription factor or not, while the funRiceGenes database (https://funricegenes.github.io/geneKeyword.table.txt, accessed on 29 August 2022) was used to search for published reports on target genes.

**3. Results**

*3.1. Differential Phenotypic Response of Parental Lines and Doubled Haploid Lines*

The parental lines (93-11, a typical *indica* ssp. and Milyang352, a typical *japonica* ssp.) and their derived population were evaluated for their phenotypic responses in different cultivation conditions. The traits studied were PN per plant, SN, grain-filling ratio, and TGW. As shown in Figure 1, 93-11 (P1) and Milyang352 (P2) had differential phenotypic responses for all traits studied. In essence, 93-11 had a higher number of panicles per plant than Milyang352 under early and late cultivation conditions in 2018 and normal cultivation in 2019. However, under normal cultivation conditions in 2021, 93-11 and Milyang352 exhibited an opposite phenotypic response (Figure 1A, Table 1). SN recorded opposite phenotypic patterns in the two different cultivation seasons of 2018 and the normal cultivation seasons in 2019 and 2021 (Figure 1B). The grain-filling ratio of 93-11 was consistently higher in all cultivation conditions compared to Milyang352 (Figure 1C). In the same way, the TGW of 93-11 was much lower under early (2018), normal (2021), and late (2018) cultivation conditions, whereas parental lines grown under normal cultivation conditions in 2019 showed an opposite pattern (Figure 1D).

**Table 1.** Grain properties of parental lines (93-11 and Milyang352) in different cropping seasons (*: significant ($p < 0.05$), **: highly significant ($p < 0.01$), ns: not significant).

| Traits | Cultivation Seasons | | | | | | | |
|---|---|---|---|---|---|---|---|---|
| | Early 2018 | | Normal 2019 | | Normal 2021 | | Late 2018 | |
| | 93-11 | Milyang 352 | 93-11 | Milyang 352 | 93-11 | Milyang 352 | 93-11 | Milyang 352 |
| Panicle number per plant | 14 ± 1.5 * | 10 ± 2.6 | 11 ± 0.3 ns | 10 ± 1.1 | 8 ± 1.8 ns | 9 ± 0.7 | 12 ± 0.2 ns | 11 ± 0.9 |
| Spikelet number per panicle | 173 ± 6.2 ** | 120 ± 3.5 | 119 ± 2.6 ** | 100 ± 4.4 | 157 ± 4.0 ** | 220 ± 5.3 | 136 ± 3.6 ** | 173 ± 3.5 |
| Grain filling (%) | 86.1 ± 0.4 ** | 57.5 ± 1.7 | 93.4 ± 2.6 ** | 78.0 ± 8.9 | 91.7 ± 8.2 ** | 66.1 ± 8.7 | 74.4 ± 7.2 ** | 61.1 ± 4.6 |
| Thousand-grain weight (g) | 19.6 ± 0.9 ** | 23.1 ± 0.9 | 25.2 ± 1.0 ** | 23.2 ± 1.2 | 22.7 ± 1.1 ** | 24.2 ± 1.6 | 20.8 ± 0.8 ns | 23.3 ± 1.7 |

When analyzing the phenotype of the mapping population relative to their parental lines (Tables 1 and 2), we observed that the majority of the DH lines had a Milyang352-like PN per plant under normal cultivation conditions in 2019. In contrast, a higher percentage of DH lines grown under normal cultivation conditions in 2021 showed a 93-11-like PN. Regarding the SN, the DH lines grown under early cultivation season in 2018 recorded a 93-11-like phenotype. As for the grain-filling ratio, a majority of the DH lines recorded a relatively low filling ratio (Milyang352-like) in all cultivation conditions, except for the

late season of 2018 (Figure 1C, Table 2). From Table 2, we could see that DH lines had a relatively high TGW under early cultivation in 2018.

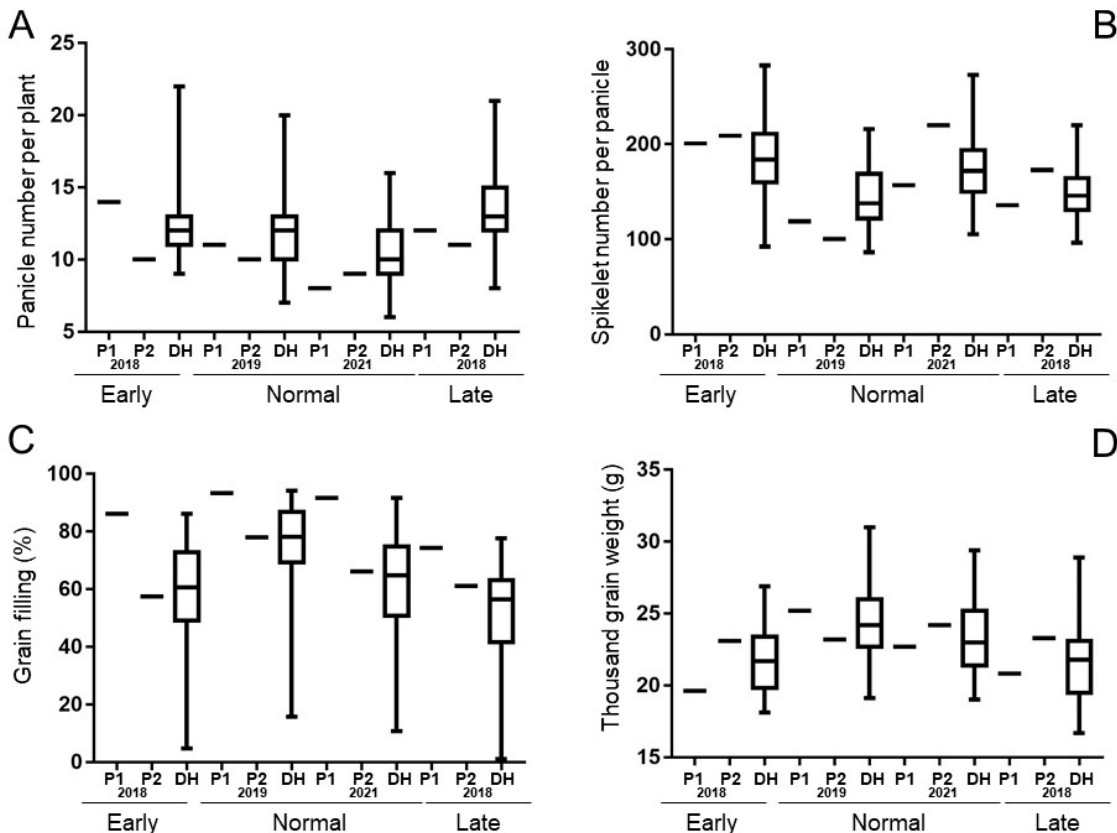

**Figure 1.** Phenotypic distribution of DH population of traits in different cropping seasons (P1: 93-11, P2: Milyang352). Phenotypes of normal season of 2021 had different distribution with normal season of 2019 except grain filling ratio. (**A**) Panicle number per plant; (**B**) Spikelet number per panicle; (**C**) Grain filling; (**D**) Thousand grain weight.

**Table 2.** Grain properties of the mapping population in different cultivation seasons. a, b, ab, c: Similarity based on *T*-test result (a: similar with 93-11, b: similar with Milyang352, ab: no difference with both parental lines, c: different with both parental lines).

| Traits | | Cultivation Seasons | | | |
|---|---|---|---|---|---|
| | | Early 2018 | Normal 2019 | Normal 2021 | Late 2018 |
| Panicle number per plant | Average ± SD | 13 ± 2.3 [ab] | 12 ± 2.3 [b] | 10 ± 1.8 [a] | 13 ± 2.3 [c] |
| | Range | 9–22 | 7–20 | 6–16 | 8–21 |
| Spikelet number per panicle | Average ± SD | 183 ± 40.2 [a] | 144 ± 28.9 [c] | 173 ± 31.4 [c] | 149 ± 26.4 [c] |
| | Range | 92–283 | 22–216 | 105–273 | 96–220 |
| Grain filling (%) | Average ± SD | 58.7 ± 16.7 [b] | 75.3 ± 13.9 [b] | 62.4 ± 16.4 [b] | 50.8 ± 26.5 [c] |
| | Range | 4.7–86.1 | 15.7–94.1 | 10.8–91.7 | 1.0–77.6 |
| Thousand-grain weight (g) | Average ± SD | 21.7 ± 2.2 [b] | 24.4 ± 2.5 [ab] | 23.3 ± 2.4 [ab] | 21.7 ± 2.7 [ab] |
| | Range | 18.1–26.9 | 19.1–31.0 | 19.0–29.4 | 16.7–28.9 |

### 3.2. Detected QTLs Associated with Grain Filling and Other Traits

To perform linkage mapping and QTL analysis, the genotype and phenotype data were employed. Among 240 SNP markers, genotype data of a total of 230 SNP markers were used for mapping, and 69 markers were deviated (Table S3). As indicated in Table 3

and Figure 2, a total of 56 QTLs were detected (all years and cropping seasons considered). Of this number, thirteen QTLs (seven under early cropping, four under normal cropping, and two under late cropping seasons of (Korean conditions) associated with grain filling (hereinafter referred to as FG) in rice were identified and mapped on five chromosomes in the present populations (Chr 3, 5, 8, 10, and 12). The *qFGL5-2*, detected in the DH population grown under late cropping season, mapped on Chr 5, had the highest logarithm of the odds (LOD) value of 15.16 and a phenotypic variation explained (PVE) of 21.66%. Other significant QTLs, *qFGE3*, *qFGE8-1* and *qFGE10*, recorded LOD and PVE values of 13.24 and 15.95%, 10.45 and 11.82%, and 8.72 and 10.42%, respectively. The findings of the GxE analysis on FG indicate that the early season of 2018 and normal season of 2019 exhibited congruent characteristics. Consequently, it is possible to discern *qFG3*, which encompasses *qFGE3* and *qFGN3*, within a particular environmental setting (Figure 3 and Table S4).

**Table 3.** Detected quantitative traits loci in different environments.

| Cultivation Season | Trait | QTL Name | Position (cM) | Left Marker | Right Marker | LOD | PVE (%) | Add | Left CI | Right CI |
|---|---|---|---|---|---|---|---|---|---|---|
| Early Season | Spikelet Number Per Panicle | *qSNE2* | 264 | KJ02_047 | KJ02_053 | 4.50 | 14.08 | −13.75 | 262.5 | 265.5 |
| | | *qSNE4* | 159 | id4009823 | cmb0432.2 | 4.23 | 14.43 | −14.86 | 152.5 | 170.5 |
| | | *qSNE9* | 60 | ae09005437 | KJ09_071 | 3.52 | 12.73 | −13.56 | 51.5 | 66.5 |
| | Panicle Number Per Plant | *qPNE2* | 268 | KJ02_053 | KJ02_057 | 5.05 | 12.25 | 0.76 | 265.5 | 272.5 |
| | | *qPNE4* | 160 | id4009823 | cmb0432.2 | 7.87 | 20.67 | 1.06 | 154.5 | 167.5 |
| | | *qPNE7* | 98 | ad07001853 | KJ07_021 | 3.95 | 9.39 | −0.66 | 93.5 | 105.5 |
| | | *qPNE9* | 42 | id9004072 | ae09005437 | 3.72 | 9.12 | 0.72 | 36.5 | 47.5 |
| | Percentage of Filled Grain | *qFGE3* | 133 | ad03013905 | ad03014175 | 13.24 | 15.95 | 8.14 | 129.5 | 135.5 |
| | | *qFGE5* | 43 | KJ05_017 | KJ05_019 | 10.88 | 12.53 | 7.23 | 39.5 | 45.5 |
| | | *qFGE8-1* | 56 | id8003584 | KJ08_053 | 10.45 | 11.82 | 7.02 | 52.5 | 58.5 |
| | | *qFGE8-2* | 86 | KJ08_085 | ae08007378 | 7.44 | 7.86 | −5.79 | 82.5 | 88.5 |
| | | *qFGE8-3* | 109 | GW8-AG | id8007764 | 4.17 | 4.18 | −4.25 | 107.5 | 109 |
| | | *qFGE10* | 70 | cmb1016.4 | wd10003790 | 8.72 | 10.42 | 7.00 | 58.5 | 75.5 |
| | | *qFGE12* | 108 | KJ12_059 | cmb1224.0 | 3.65 | 3.57 | −3.81 | 103.5 | 112.5 |
| | Thousand Grain Weight | *qTGWE1-1* | 230 | KJ01_121 | KJ01_125 | 4.90 | 3.17 | −0.63 | 221.5 | 238.5 |
| | | *qTGWE1-2* | 285 | P1193 | KJ01_129 | 4.32 | 2.77 | −0.44 | 275.5 | 285.5 |
| | | *qTGWE2* | 252 | KJ02_039 | KJ02_043 | 8.37 | 8.12 | −0.75 | 238.5 | 262.5 |
| | | *qTGWE3-1* | 30 | ad03000001 | KJ03_007 | 10.60 | 7.41 | 0.71 | 22.5 | 32.5 |
| | | *qTGWE3-2* | 183 | KJ03_069 | Hd6-AT | 10.58 | 7.64 | −0.73 | 172.5 | 187.5 |
| | | *qTGWE5* | 38 | id5002497 | KJ05_013 | 19.80 | 16.94 | −1.08 | 32.5 | 39.5 |
| | | *qTGWE7* | 173 | KJ07_067 | cmb0723.0 | 9.24 | 6.25 | 0.68 | 171.5 | 175.5 |
| | | *qTGWE8* | 109 | GW8-AG | id8007764 | 10.88 | 7.58 | 0.74 | 107.5 | 109 |
| | | *qTGWE10* | 130 | KJ10_049 | id10007384 | 12.66 | 9.33 | −0.79 | 127.5 | 130 |
| | | *qTGWE12-1* | 14 | cmb1202.4 | KJ12_007 | 9.75 | 6.69 | 0.67 | 9.5 | 23.5 |
| | | *qTGWE12-2* | 127 | cmb1226.0 | id12010130 | 6.26 | 3.95 | −0.52 | 122.5 | 131 |
| Normal Season (2019) | Spikelet Number Per Panicle | *qSNN4* | 192 | KJ04_093 | id4012434 | 4.51 | 10.72 | −12.05 | 186.5 | 192 |
| | | *qSNN8* | 88 | ae08007378 | id8006751 | 5.77 | 14.29 | −13.20 | 82.5 | 91.5 |
| | | *qSNN12* | 56 | id12003700 | KJ12_041 | 6.09 | 14.87 | −13.73 | 51.5 | 60.5 |
| | Panicle Number Per Plant | *qPNN2* | 267 | KJ02_053 | KJ02_057 | 4.22 | 11.83 | 0.73 | 265.5 | 272.5 |
| | | *qPNN4* | 162 | cmb0432.2 | cmb0434.1 | 4.25 | 12.26 | 0.79 | 154.5 | 169.5 |
| | | *qPNN5* | 102 | ad05008445 | id5010886 | 3.52 | 11.47 | −0.77 | 82.5 | 112.5 |
| | | *qPNN7* | 106 | KJ07_021 | id7001155 | 4.99 | 14.68 | −0.81 | 98.5 | 114.5 |

**Table 3.** *Cont.*

| Cultivation Season | Trait | QTL Name | Position (cM) | Left Marker | Right Marker | LOD | PVE (%) | Add | Left CI | Right CI |
|---|---|---|---|---|---|---|---|---|---|---|
| | Percentage of Filled Grain | *qFGN3* | 133 | ad03013905 | ad03014175 | 5.17 | 5.97 | 5.11 | 128.5 | 138.5 |
| | | *qFGN5-1* | 38 | id5002497 | KJ05_013 | 11.05 | 14.40 | 7.86 | 31.5 | 39.5 |
| | | *qFGN5-2* | 130 | KJ05_063 | cmb0526.3 | 6.33 | 7.41 | −6.95 | 127.5 | 130.5 |
| | | *qFGN5-3* | 139 | KJ05_071 | id5014265 | 11.48 | 15.05 | 9.90 | 138.5 | 140.5 |
| | Thousand Grain Weight | *qTGWN2* | 290 | ae02004877 | cmb0232.7 | 7.53 | 15.18 | −1.22 | 284.5 | 296.5 |
| | | *qTGWN8* | 109 | GW8-AG | id8007764 | 6.55 | 12.48 | 1.10 | 107.5 | 109 |
| | | *qTGWN12* | 9 | id12000076 | cmb1202.4 | 6.33 | 12.92 | 1.08 | 2.5 | 13.5 |
| Late Season | Spikelet Number Per Panicle | *qSNL3* | 82 | id3005168 | ah03001094 | 4.67 | 8.07 | −8.16 | 74.5 | 91.5 |
| | | *qSNL4* | 150 | ah04001252 | id4009823 | 12.51 | 23.17 | −14.62 | 146.5 | 154.5 |
| | | *qSNL5* | 150 | id5014265 | cmb0529.7 | 3.29 | 5.07 | −7.17 | 149.5 | 150 |
| | | *qSNL7* | 158 | id7003072 | KJ07_067 | 11.93 | 22.70 | −14.07 | 154.5 | 163.5 |
| | Panicle Number Per Plant | *qPNL2* | 254 | KJ02_039 | KJ02_043 | 4.26 | 13.21 | 0.83 | 227.5 | 262.5 |
| | | *qPNL3* | 196 | Hd6-AT | id3015453 | 5.42 | 12.65 | 0.83 | 193.5 | 209.5 |
| | | *qPNL4* | 147 | ah04001252 | id4009823 | 7.81 | 19.81 | 1.09 | 140.5 | 149.5 |
| | | *qPNL7* | 105 | KJ07_021 | id7001155 | 3.07 | 7.19 | −0.61 | 98.5 | 108.5 |
| | Percentage of Filled Grain | *qFGL5-1* | 130 | KJ05_063 | cmb0526.3 | 6.79 | 8.08 | −9.99 | 127.5 | 130.5 |
| | | *qFGL5-2* | 139 | KJ05_071 | id5014265 | 15.16 | 21.66 | 16.38 | 138.5 | 141.5 |
| | Thousand Grain Weight | *qTGWL2* | 245 | KJ02_039 | KJ02_043 | 4.83 | 12.30 | −0.98 | 221.5 | 262.5 |
| | | *qTGWL3-1* | 53 | id3003462 | id3005168 | 6.19 | 8.76 | 0.83 | 47.5 | 63.5 |
| | | *qTGWL3-2* | 230 | ah03002520 | cmb0336.5 | 6.59 | 8.57 | −0.82 | 221.5 | 233 |
| | | *qTGWL5* | 39 | KJ05_013 | KJ05_017 | 10.89 | 15.44 | −1.10 | 38.5 | 41.5 |
| | | *qTGWL7-1* | 0 | cmb0700.1 | KJ07_011 | 4.72 | 5.91 | −0.68 | 0.0 | 21.5 |
| | | *qTGWL7-2* | 160 | id7003072 | KJ07_067 | 5.80 | 8.36 | 0.84 | 152.5 | 166.5 |
| | | *qTGWL10* | 130 | KJ10_049 | id10007384 | 7.84 | 10.66 | −0.91 | 123.5 | 130 |

(a) Cultivation seasons of the mapping population: early cultivation, standard rice cultivation, late cultivation; (b) rice traits used for the QTL analysis; (c) chromosome number; (d) QTL associated with traits studied; (e) absolute position of detected QTLs; (f) left flanking markers; (g) right flanking markers; (h) logarithm of the odds (LOD) scores; (i) phenotypic variation explained (PVE) by the QTLs expressed in percentage; (j) additive effects: the negative value indicates that the allele from Milyang352 increased the corresponding trait value, while the position value denotes that the allele from 93-11 increased the trait value.

It is also found that *qFGE8-3* and QTLs related to TGW (*qTGWE8* and *qTGWN8*) are flanked by the same markers (GW8-1AG and id8007764) but are mapped at different positions of the same chromosome (Chr 8). Similarly, another set of two adjacent QTLs, *qFGE5* and *qFGN5-1*, *qFGN5-2* and *qFGL5-1*, and *qFGN5-3* and *qFGL5-2*, were mapped on Chr 5 and here proposed to govern grain filling in rice.

Data in Table 3 also indicate that QTLs *qSNL5* is adjacent to *qFGN5-3* and *qFGL5-2* on chromosome 5 (27.80 Mbp) and flanked by KJ05_063 and cmb0529.7 marker. When comparing QTLs detected in different environments (Figure 2 and Figure S3), we could see that *qFGE5* is adjacent to *qTGWE5* and *qTGWL5* controlling TGW under early and late cultivation conditions (flanking markers: id5002497 and KJ05_019).

Furthermore, ten QTLs controlling the SN were mapped on eight chromosomes of rice (Chr 2-5, 7-9, 12), of which number *qSNL4* and *qSNL7* recorded the highest LOD scores and PVE percentage.

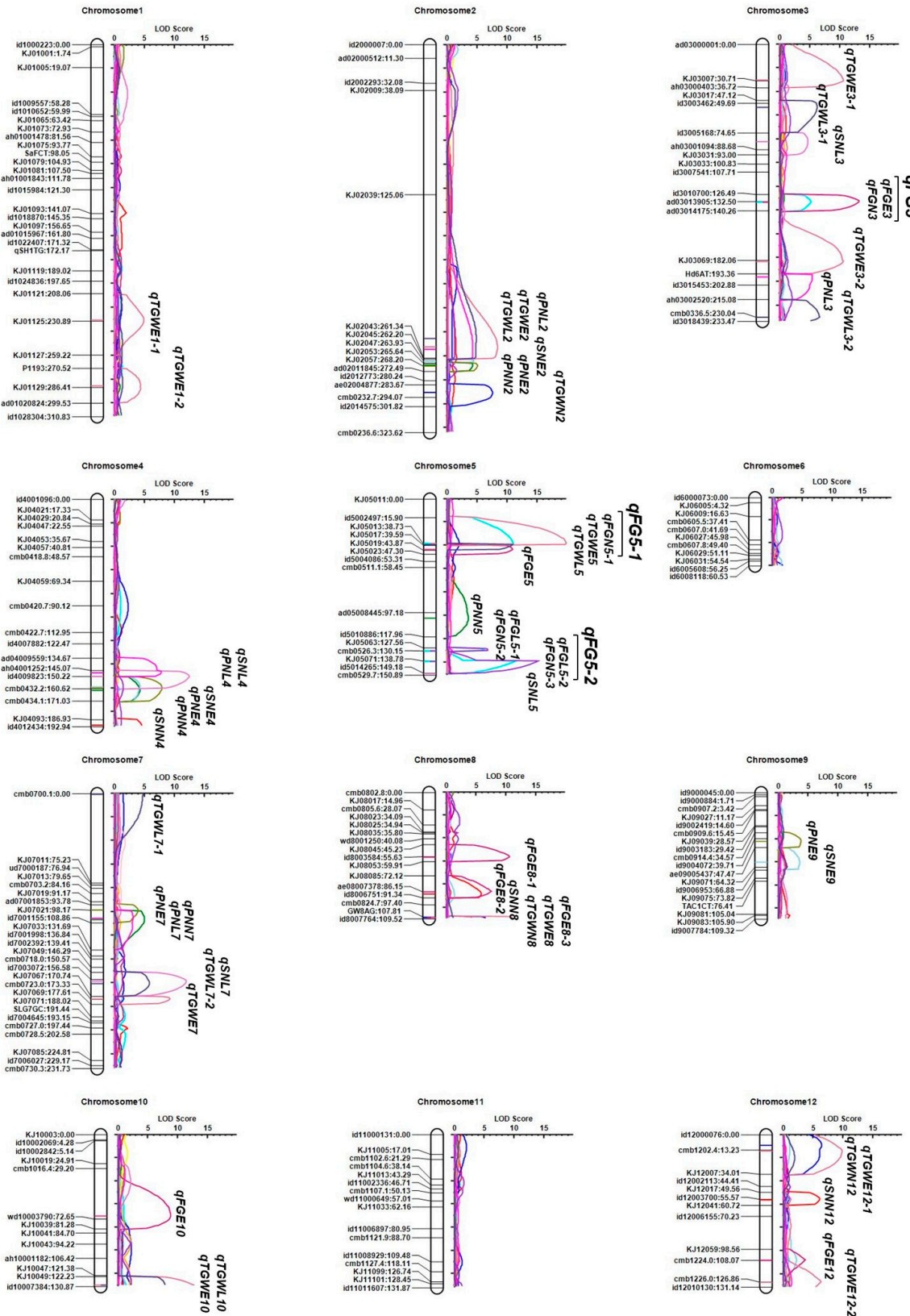

**Figure 2.** Linkage map of grain traits-related QTLs. *qFGE3* and *qFGN3* are repeatedly detected on close region of Chromosome 3 in different cultivation seasons.

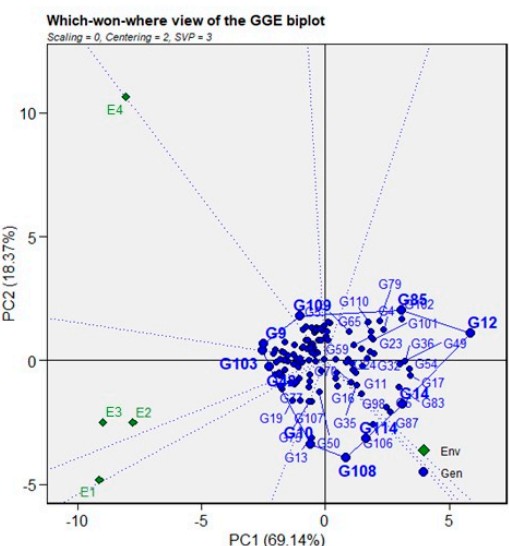

**Figure 3.** GxE analysis result of FG (E1: Early season, E2: Normal season of 2019, E3: Normal season of 2021, E4: Late season of 2018). The result indicates *qFG3* is detected under certain Environment.

Moreover, PN and TGW are widely recognized as important rice traits that reflect rice productivity. Here, data in Table 3 show that a total of twelve QTLs associated with the control of PN were detected (all cultivation conditions). Among them, two set of QTLs, *qPNE2* and *qPNN2,* and *qPNN7* and *qPNN7,* mapped on Chr 2 and Chr 7 of the mapping population (Table 3), were flanked by the same markers (KJ02_053 and KJ02_057, KJ07_021 and id7001155), respectively. Similarly, twenty-one QTLs associated with TGW were mapped on eight chromosomes (Chr 1–3, 5, 7, 8, 10, and 12). Interestingly, *qTGWE5* and *qTGWL5* flanked by KJ05_013 and KJ05_017 KASP markers coincided with a major QTL (*qGW-5-1/qGT-51/qGLWR-5-1*, 24 cM) recently proposed to govern grain shape (width, thickness, and grain length–width ratio) in rice [32]. Based on this observation, we could say that TGW and grain shape traits would be controlled by the same genetic locus, implying that both traits could be tightly related.

### 3.3. Genes Harbored by qFG3, qFG5-1 and qFG5-2 QTLs Associated with Grain Filling in Rice

To calculate recombination, 210 Markers from 230 genotyping data of polymorphic SNP markers were mapped and grouped into 12 linkage groups covering 2116.6 cM (Table S1). Twelve QTLs associated with grain filling (FG) were detected in different cultivation periods. We were particularly interested in unveiling the identity of genes located in QTLs for grain filling under early cultivation conditions. In this regard, among the seven QTLs for FG in early cultivation, we identified two detected by our study for deep mining. The other remaining five QTLs have already been reported by previous studies [32]. As Kang et al. (2020) reported, those QTLs are found to be associated with QTLs related to grain shape or weight. Thus, a deep search of gene ontology fetching from several genome annotation databases was freely available online. As shown in Table S6, the result of this search reveals the biological processes and the molecular functions characterizing genes harbored by *qFG3*. In essence, genes encoding a tubulin/FtsZ domain-containing protein, DNA-directed RNA polymerase II subunit RPB2, β-subunit, Chaperone protein DnaJ [33,34], are associated with cell division, reproduction and embryo development, and flower development, respectively. The *qFG3* also harbors genes related to cell differentiation and cellular component organization, including *OsSh1* [35–37], *SWI1* [38–44], and Os03g45260, similar to the Arabidopsis Membrin 11 (*AtMEMB11*).

In the same genetic locus, genes annotated as being involved in ion transport or ion binding are found, including Os03g44484, Os03g44630, Os03g45370, Os03g44636, and Os03g44660. Another set of genes located in the *qFG3* region, said to be involved in the

transcriptional regulation of specific genes or translation, are Os03g44540, Os03g44780, Os03g44820 [45], Os03g44900 [46], Os03g45400, and Os03g45450.

Likewise, a category of genes involved in the metabolic process (Os03g44500 and Os03g45390, involved in carbohydrate metabolism), catalytic process (Os03g45194) [47–49], and biosynthetic process (Os03g45320 and Os03g45420) were identified [50]. We could, additionally, find two genes encoding a cytochrome P450 protein (Os03g44740 and Os03g45619). The QTL *qFG3* equally harbors genes belonging to the family of proteins of unknown function (DUF266, Os03g44580) [51,52], encoding an IQ calmodulin-binding motif domain (Os03g44610) [53–55], a peptidase S54 rhomboid domain-containing protein (*OsRhmbd9*, Os03g44830, having a serine-type endopeptidase activity) [56], a C2 domain-containing protein (Os03g44890, having a transferase activity) [57], a protease inhibitor/seed storage/LTP protein precursor (*LTPL91*, Os03g44950) [58,59], or encoding an ankyrin repeat domain-containing protein 50 (Os03g45920) [60,61].

From the above phenotype of the mapping population, we were interested in assessing the eventual effect of *qFG3* on different traits evaluated. From data in Figure S4A,C and Table 3, it is confirmed that alleles from 93-11 had significantly enhanced the observed phenotypic variation of SN and grain-filling ratio in the early season. However, for SN, the alleles from Milyang352 contributed to the observed phenotypic variation (Figure S4B, Table 3). Unlike other traits, a non-significant difference was found between the effects of alleles from 93-11 and Milyang352 for TGW (Figure S4D).

Intriguingly, the majority of genes included in QTLs *qFG3*, *qFG5-1* and *qFG5-2*, in addition to their involvement in developmental and cell differentiation processes, are proposed to be involved in the adaptive response mechanism toward abiotic stress or the oxidation-reductase process (Table S5 and S6).

## 4. Discussion

### 4.1. Differential Grain-Filling Ratio between Parents and DH Lines

Rice genotypes exhibiting a high grain-filling ratio are generally preferred to those with an opposite phenotype, mainly because a high grain filling ratio is expected to result in high yield. Sink strength and grain filling are two of the major determinants of rice yield, with carbohydrates from photosynthetic assimilates and non-structural carbohydrates (NSC) present in culms and leaf sheaths playing important roles [62]. However, Zhang et al. [37] suggested that, although grain filling in rice largely consists of starch accumulation (about 90% of the dry weight of grains), carbohydrate supply may not be the only limiting factor for poor grain filling, especially for inferior spikelets as compared with superior ones.

Here, we recorded a differential grain-filling ratio of the parental lines and their derived mapping population cultivated in different seasons. The majority of the DH lines exhibited a low grain-filling ratio similar to that observed in Milyang352 (P2, *japonica*) in all cultivation seasons, while only a few DH lines showed a P1-like grain-filling pattern. The highest grain-filling ratio (75.3%) was obtained in the normal rice cultivation season in Korea in 2019.

In their study, Sekhar et al. [63] observed that grain filling in rice was significantly influenced by factors such as spikelet fertility, coupled with panicle compactness and ethylene production. However, our data did not detect any strong correlation between grain-filling ratio and other traits in all cultivation seasons (Figure S2).

### 4.2. A Novel QTL Associated with Grain Filling Is Identified

Several QTLs reported to control grain filling have been found and mapped to several chromosomes of rice. Some of these QTLs are also associated with the control of other grain traits, including grain shape and size, TGW (*GW2* [64], *qSW5/GW5*, *GW7/GL7*, and *GW8*, *GS3*, 5, and 6 [65–76]), spikelet fertility (*qSFP1.1*, *qSFP3.1*, *qSFP6.1*, *qSFP8.1*), panicle compactness (*qIGS3.2*, *qIGS4.1*), and ethylene production (*qETH1.2*, *qETH3.1*, *qETH4.1*, *4.2*, *qETH6.1*, *6.2*). Our study identified seven QTLs associated with rice grain filling, of which number *qFG3* is novel, and was detected in both early and normal cultivation

seasons. *qFG3* is therefore regarded as a stable QTL, and could serve as a useful source for downstream breeding.

We could see that *qFG3* harbors genes associated with cell division, elongation, and differentiation, including a member of the tubulin superfamily (tubulin/FtsZ, major components of the cytoskeleton of eukaryotes). In plants, tubulin contributes to the formation of microtubules [77] that, in turn, control cell division, growth polarity, intracellular trafficking and communications, cell-wall deposition, and cell morphogenesis [78–80]. As primary components of microtubules [77], plant tubulins (α, β, and γ) provide structural support for overall cell shape as well as the transport and positioning of organelles [78]. Other genes encoding α-tubulin, *Srs5* (*small and round seed 5*) [81] and β-tubulin (*OsTUB8*) [82] are associated with seed formation, elongation or size [83].

The gene ontology search of *qFG3*-related genes showed that Os03g44420 (a tubulin/FtsZ domain-containing protein) or glutamyl-tRNA synthetase amidotransferase C subunit [44] are involved in cell division and differentiation, or ion transport. Another set of two genes encoding β-catenin repeat protein (Os03g45420) or an MYB-like DNA-binding domain-containing protein (Os03g45197) is proposed to be involved in the regulation of chlorophyll and ABA biosynthesis processes or photosynthesis. A relationship between grain filling, yield and chlorophyll content has been established. Reports suggest that rice grain yield is highly dependent on the photosynthetic assimilation of leaves during the grain-filling stage, where about 60–80% of the nutrients required for grain filling are contributed by the photosynthesis of source leaves during the same period. In contrast, Chen et al. [84] investigated the genetic diversity of six rice cultivars for their grain-filling rate and revealed that the contribution of photosynthesis to rice grain filling would be genotype-dependent.

*4.3. QTL qFG3 Harbors Genes with Transcription Factor Activity*

In all biological systems, gene regulation is governed through a combinational action of multiple regulatory proteins, including transcription factors (TFs) [85]. As shown in Table S6, four genes encoding TFs were found within the genetic region covered by *qFG3*. These genes encode Nf-Y-A subunit (Os03g44540), CCR4-NOT (Os03g44900), antitermination NusB domain-containing protein (Os03g45400), WRKY60 (Os03g45450), and YABBY domain-containing protein (*OsSh1*, shattering1) [35]. Several studies have shown that genes containing the YABBY domain functionally interact with other genes or biological compounds to regulate plant growth and development [86]. However, Dai et al. [36] demonstrated that a *WUSCHEL-LIKE HOMEOBOX* gene acts as a negative regulator of a *YABBY* gene required for rice leaf development.

*4.4. qFG3-Related Genes Are Associated with Abiotic Stress Response Mechanisms*

Plant growth and development are often favored under normal conditions, while slow and impaired growth is commonly observed when plants experience stress conditions. In this study, a set of genes located in the *qFG3* QTL region is proposed to be involved in the adaptive response mechanisms (Table S6). It is well established that reactive oxygen (ROS) and nitrogen (RNS) species play important roles in maintaining regular plant growth under normal conditions, such as seed development and germination, shoot and flower development, etc. [87]. ROS and RNS are by-products of aerobic metabolism generated in various cellular compartments, including chloroplasts [88,89], mitochondria [90,91], and peroxysomes [92,93]. Within this perspective, *OsIPMPDH* (Os03g45320), having an oxidoreductase activity and magnesium ion binding, and those encoding cytochrome P450 protein, coupled with Os03g44620 andOs03g44630, can be targets to investigate the relationship between grain filling and stress response in rice.

**5. Conclusions**

Grain filling is a determinant trait for the yield of cereal crop species, such as rice. This study identified QTLs associated with grain filling in rice under different cultivation

seasons. Among them, *qFG3*, *qFG5-1*, and *qFG5-2* were significant, of which *qFG3* recorded a high LOD score. Genes harbored by *qFG3* are associated with cell division, embryo and post-embryo development, photosynthesis, and starch synthesis, among others. Genes such as MYB, WRKY60, and *OsSh1* encoding TFs, β-catenin, and the tubulin FtsZ have interesting predicted molecular functions. Another set of genes are proposed to be involved in abiotic stress signaling or adaptive response mechanisms. *qFG3* is a QTL that can be detected in a specific environment. It can serve as a target QTL region for downstream breeding, including the fine mapping and functional characterization of putative candidate genes for their roles during grain filling in rice.

**Supplementary Materials:** The following supporting information can be downloaded at: https://www.mdpi.com/article/10.3390/agronomy13030912/s1, Figure S1: Example of allele discrimination plot of polymorphic SNP marker (KJ11_13, Table S2).; Figure S2: Correlation between grain properties and yield component with normal fertilization level (y: FG, x: SN or PN or TGW, SN: spikelet number per panicle, PN: panicle number per plant, TGW: thousand-grain weight (g), FG: percentage of filled grain (%), E: early season, N: normal season, L: late season, R2: coefficient of determination).; Figure S3: Weather conditions prevailing during 2018, 2019, and 2021 cultivation seasons.; Figure S4: The distribution of grain properties of the two groups differed by the genotype of *qFG3* in different fertilization levels and cropping seasons.; Table S1: cM of each chromosome.; Table S2: List of SNP markers.; Table S3: Information on the physical distance between SNP markers.; Table S4: Distribution of flowering time of DH population according to the genotype of *qFG3*.; Table S5: List of candidate genes; Table S6: Putative candidate genes located within *qFG3*.

**Author Contributions:** Conceptualization, J.-H.L.; methodology, J.-W.K., J.-H.L. and N.R.K.; software, S.-M.L. and N.R.K.; validation, J.-H.L.; formal analysis, S.-M.L., N.R.K., Y.K., J.-K.C. and H.P.; investigation, S.-M.L., J.-W.K. and J.S.; resources, J.-W.K., J.S. and J.-H.L.; data curation, S.-M.L. and N.R.K.; writing—original draft preparation, S.-M.L. and N.R.K.; writing—review and editing, J.-H.L.; visualization, J.-H.L.; supervision, J.-H.L., H.-J.K. and K.-W.O.; project administration and funding acquisition, J.-H.L. All authors have read and agreed to the published version of the manuscript.

**Funding:** This research was funded by the Rural Development Administration, Republic of Korea, grant number PJ01428201.

**Data Availability Statement:** Not applicable.

**Acknowledgments:** We are grateful for the support from the Rural Development Administration (RDA) and Crop Molecular Breeding lab in Seoul National University (SNU), Republic of Korea.

**Conflicts of Interest:** The authors declare no conflict of interest.

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
