# Peer review of "Identifying QTLs Related to Grain Filling Using a Doubled Haploid Rice (Oryza sativa L.) Population"

_agronomy, doi:10.3390/agronomy13030912_

Round 1

Reviewer 1 Report (Previous Reviewer 1)

This manuscript, agronomy-2268471 (revised one of agronomy-2177101), was not improved sufficiently. It still involved incorrect, meaningless, and lengthy contents as the previous one. I believe this manuscript still did not attain the level of acceptance for the publication in Agronomy, and should be rejected. If the authors will delete completely the contents in the sections 3.3, 3.4, 4.1, 4.3, 4.4, 4.5, Conclusion, and related Tables and Figures, this manuscript may be accepted in some local journal, but not be suitable for international journals like Agronomy.

L40-41: Delete “, which is closely associated with the source ability”, or revise “, which is closely associated with the sink and source abilities”.

L88: row length (?)

L100: Please show us the specific gravity of NaCl solution of 130 g/L.

L149-150: Delete the sentence (“Finally, … used for mapping.”). This is duplicated to L145.

Table 1: It is better to be understood that “*” and “**” are attached to larger means.

Table 2: What was the means without letters? (For example, 12±2.3 for panicle number per plant in Normal 2019.)

L252-253: The authors described that under early cultivation spikelet number was negatively correlated with grain filling ratio. Since R2 = 0.01, r should be -0.1. From the 115 degree of freedom, this coefficient was not significantly different from 0 (r (0.05) = 0.182, r (0.01) = 0.238). Thus, it cannot be concluded generally that two traits were correlated. Rather, grain filling rate was correlated significantly and negatively with PNE (r = 0.232, P<0.05) and TGWN21 (r = 0.190, P<0.05). Therefore, this section and also section 4.1 were nonsense.

L255-256: I can never understand this sentence. What is the “unit increase” and “by the same measure”?

L257-272: I can hardly understand these paragraphs. I believe the differences in weather conditions among growing seasons cannot contribute directly to the differences in the performance of DH lines particularly in this experiment. I strongly recommend to authors to delete completely these paragraphs and Figure S3.

L274-314: These paragraphs were merely a meaningless list of annotated loci around the detected QTLs for grain filling rate. Why can the authors claim that qFG3, for example, harbored such many candidate loci? As the previous manuscript, sections 4.1, 4.3, 4.4 and 4.5 were nonsense, and should be deleted.

L479: This Conclusion involved incorrect descriptions (“strong” positive correlation (R2 = 0.01)), nonsense discussion, and many duplications, and should be deleted.

Author Response

We appreciate your valuable comments, and our answer is in the attachment. Please see the word file. Thank you.

Reviewer 2 Report (Previous Reviewer 2)

already revised

Author Response

Thank you for your constant attention for our manuscript.

Reviewer 3 Report (Previous Reviewer 3)

see my suggestions in attached pdf

Author Response

We are very thankful for your comments. Please see the attachment for the answer for your comments.

Round 2

Reviewer 3 Report (Previous Reviewer 3)

none of query raised by me in V0 not resolved article can not be accepted in current form

Author Response

We appreciate your valuable comments. please check the attachment to see our response

This manuscript is a resubmission of an earlier submission. The following is a list of the peer review reports and author responses from that submission.

Round 1

Reviewer 1 Report

This manuscript, agronomy-2177101, studied about the QTLs for grain filling (percentage of filled grains) and other related traits in rice. The objective of this study, in itself, is very important for rice breeding toward higher-yield. The authors demonstrated finally that they identified two novel QTLs for percentage of filled grains, and these QTLs would concern several important physiological events in plants. However, I strongly believe that these conclusions and most of the logics in the text were based on misunderstand and scientific immatureness of authors, and that this manuscript did not, and also will not, attain the level of acceptance for publication in Agronomy, even after major revision. I conclude that this manuscript should be rejected for publication. The reasons are as follows.

1. The authors identified 2 or 3 major QTLs for the percentage of filled grain (qFG), and listed up 30 (for qFGE3) and 37 (for qFG10E) registered loci which are located within the estimated regions of these QTLs. Finally, the authors concluded that several loci of them were identified to be associated with grain filling in rice, after searching the reported functions of these loci. This is too immature and too crude to be concluded. If the authors want to demonstrate the function of novel QTLs, they should first conduct very fine mapping of the QTL with more markers and concentrate the QTL region to, at least, within several hundred kbp. And they should search SNPs in the several (not 30 or more) candidate loci in this region between the parent types, and should confirm the functional differences between the different alleles on these loci by comparing the FG between parental SNP types. The present study never conducted these kinds of research. The authors may perhaps claim that this was done in Figure 2. However, this was for only the genotypes of a single marker linked qFG3E. Therefore, all of the descriptions in section 3.4, Discussion, Conclusion, Table 5 and Table S1, and also Abstract, are nonsense.

2. The mapping population (117 DH from 93-11 and Milyang 352) and markers used (240 KASP markers) in this study were the same as those of Kang et al. (Agronomy 10: 1532, 2020). The present authors should cite this information. In the same year and season, however, the thousand grain weight (TGWE and TGWL) showed apparently different values of the parents between two studies (Figure 1D in the present study and Figure 3E in Kang et al.). Why? Also in the present study, Figure 3 showed that some DH lines had very fewer numbers of spikelets per panicle and also showed very low FG. One line had nearly no spikelets per panicle (Figure 2A and Figure 3). This is incredible. If the authors conducted QTL analysis and others including such extraordinary DH lines (those with less than 50 spikelets per panicle, for example), the results should involve serious problems, and not to be acceptable. I, therefore, cannot help being suspicious for the material handling of the authors and for the results derived from it.

3. The authors demonstrated that in early season of 2018 the FG was significantly and positively associated with spikelet number per panicle. From Figure 3, this was probably the same if the extraordinary lines, as above, was excluded. This association is very novel and strange from the previously obtained trend, that is, grain filling and spikelet number is negatively associated. However, from Figure 2A and 2C, the DH lines showing 93-11 marker types were significantly higher FG in 2021, but they were significantly lower number of spikelets in the same year. Why this novel association was only detected in early season of 2018?

4. This manuscript is quite lengthy. Particularly for the Results section, the authors only repeated the results in Tables and Figures, one by one. The authors should make very concise article as far as possible. Please make manuscript or article for readers, and not for authors themselves. Also, the quality of English in this manuscript was poor.

Minor problems to be corrected are listed below:

L18-19: If so, why did the authors measure the traits for source activity? The only examined the traits for sink capacity.

L28: …were identified, that is (or :) qFGE3 ( ) and qfGE10 ( ). Delete “are novel QTLs”.

L28, 29: “Logarithm of the odds” can be omitted. This abbreviation (LOD) may be obvious for readers.

L32-34: This sentence is ambiguous. Probably, “…, and embryo, post-embryo and flower development;(?) and starch synthesis.”

L35: found => detected (?)

L49: Is it necessary in this context that “The breakthrough is responsible for improved rice yield”? I think this can be deleted.

L68: indica allele at a QTL

L76: indica allele at qLIA3

L83, 85: allele at LSCHL4

L92: , of which two QTLs (qFGE3 and qFGE10) with …

L104: Milyang (?)

L105 and others: 4 m, 15 cm, 30 cm, 45 days, etc.

L116-126: I think these descriptions for the FG measurement can be simplified. For example, “the filled grains in this study were defied as grains sinking into the NaCl solution of 130 g L-1 (please show the specific gravity of this solution). The filled grain ratio (FG) was calculated as … “

L128: average filled (?) one-grain weight x 1000 (?)   Is (n) necessary?

L132: Is this sentence “The phenotype…” necessary? This is obvious.

L141: What is “e-tube”?

L146-154: Too many “and”. Revise as L155-L164.

L176: Probably, “… to perform linkage mapping and QTL analysis with … were selected).”

L199: … typical japonica ssp.) and their derived …

L227: As for the grain filling ratio, a majority of the DH lines …

L234-242: This paragraph and Figure 2 should be removed to after L449. Because the readers cannot understand what is the qFGE3.

L261: plant => panicle

L265: Delete “Rice”. Too obvious.

L266: The first sentence is too obvious for the readers, and should be deleted.

L268: Of these,

L271: in rice => in the present populations

L277, 278: I cannot understand “in (In?) the same other” and “A close look at data”.

L332-L337: These sentences should be deleted, because they are too obvious.

L340: Please show us the other correlation data clearly.

L344-362: Nobody can understand why these paragraphs were appeared here. These are the explanations of meteorological profiles of the growing years and seasons in the present study. However, the title of this section was “Correlation analysis …”. No relations.

L370: I, and also most of the readers, cannot understand why the authors search only qFGs in the early season. For example, qFG8 in the normal season (2021) showed the highest LOD and PVE among qFGs.

L372: Please show us the literature for the already reported previous studies. These studies were not cited nowhere in the text.

L373: Where is the verb of this sentence?

L509: Figure 2 reveals only the effects of qFG3, not qFG5. Therefore, this sentence is incorrect.

L696: What is the several letters after the authors’ name (J.A.S, J.F.C.R. and so on)? Delete all of them.

Table 1 and 5: Please follow the general rule for making tables to delete the vertical lines in these figures.

Author Response

Dear reviewer,

We appreciate your kind advice.

Please see the attachment for our response and revised manuscript.

Sincerely, So-Myeong Lee

Reviewer 2 Report

I have following suggestions and comments

1. the introduction section is too short and authors fails to provide necessary literature related to rice and rice yield, also fail to discuss the signaficance of grain filling trait in improving rice yield. please revise the introduction.

2. line 88 need to be corrected it should not be genetic factors. I will suggest to use genetic regions/genomic regions

3. in material method section 2.1, the authors fails to providing breeding design, number of replications, plot size, row and plant spacing and other culture practices. I will recommend to add all the important informations of culture practices

4. in material method section 2.3, no need to add all the DNA isolation process

5. please add additional information in Table 1 in footnotes

6. Please explain the y, R2 and FG (%) in figure 3

7. Table 5 should be move to additional files/supplimentry data

8. please remove line 444 to 448 from results section and adjust somewhere in discussion section

9. provide a refrence at line number 181 at 3 threshold 3.0.

Author Response

(The authors gave the same response as above.)

Reviewer 3 Report

See my suggestions in attached pdf

Author Response

(The authors gave the same response as above.)

Round 2

Reviewer 3 Report

Some queries raised in last version like correlation methodology, linkage map result and its discussion not included in revision. Food study please include for acceptance and for better understanding of readers.

Author Response

Dear reviewer,

We appreciate your kind advice.

Please see the attachment for our response and revised manuscript.
